# Radiomics, a Promising New Discipline: Example of Hepatocellular Carcinoma

**DOI:** 10.3390/diagnostics13071303

**Published:** 2023-03-30

**Authors:** Thomas Lévi-Strauss, Bettina Tortorici, Olivier Lopez, Philippe Viau, Dann J. Ouizeman, Baptiste Schall, Xavier Adhoute, Olivier Humbert, Patrick Chevallier, Philippe Gual, Lionel Fillatre, Rodolphe Anty

**Affiliations:** 1Hepatology Unit, University Hospital of Nice, 151 Route de Saint Antoine de Ginestière, 06200 Nice, France; levi-strauss.t@chu-nice.fr (T.L.-S.);; 2Department of Diagnosis and Interventional Imaging, University Hospital of Nice, 151 Route de Saint Antoine de Ginestière, 06200 Nice, France; 3Department of Nuclear Medicine, University Hospital of Nice, 151 Route de Saint Antoine de Ginestière, 06200 Nice, France; 4CN3S, I3S, Université Côte d’Azur, 06000 Nice, France; 5Saint Joseph Hospital, 26 Bd de Louvain, 13008 Marseille, France; 6Centre Antoine-Lacassagne, Department of Nuclear Medicine, 33 Av. de Valombrose, 06100 Nice, France; 7TIRO-UMR E 4320, Université Côte d’Azur, 06000 Nice, France; 8INSERM, U1065, C3M, Université Côte d’Azur, 06000 Nice, France

**Keywords:** radiomics, artificial intelligence, precision medicine, hepatocellular carcinoma

## Abstract

Radiomics is a discipline that involves studying medical images through their digital data. Using “artificial intelligence” algorithms, radiomics utilizes quantitative and high-throughput analysis of an image’s textural richness to obtain relevant information for clinicians, from diagnosis assistance to therapeutic guidance. Exploitation of these data could allow for a more detailed characterization of each phenotype, for each patient, making radiomics a new biomarker of interest, highly promising in the era of precision medicine. Moreover, radiomics is non-invasive, cost-effective, and easily reproducible in time. In the field of oncology, it performs an analysis of the entire tumor, which is impossible with a single biopsy but is essential for understanding the tumor’s heterogeneity and is known to be closely related to prognosis. However, current results are sometimes less accurate than expected and often require the addition of non-radiomics data to create a performing model. To highlight the strengths and weaknesses of this new technology, we take the example of hepatocellular carcinoma and show how radiomics could facilitate its diagnosis in difficult cases, predict certain histological features, and estimate treatment response, whether medical or surgical.

## 1. Introduction

At the time of precision and personalized medicine, new biomarkers are needed to better understand disease profiles and improve diagnosis accuracy and precise prognosis. Radiomics is an emerging field that uses artificial intelligence technology to extract, analyze, and interpret medical imaging data. Taking the example of hepatocellular carcinoma (HCC), we will show its promise, results, and limits. First, we will define radiomics, and then we will see, with a selection of studies, how a radiomics analysis of data from Computed Tomography (CT), Magnetic Resonance Imaging (MRI), and Positron emission tomography–computed tomography (PET-CT) is applied to HCC.

## 2. What Is Radiomics?

Artificial intelligence (AI) is currently experiencing major developments, particularly in healthcare. AI refers to the use of computer algorithms to mimic the cognitive functions of the human brain, i.e., tasks that require memory, judgment, and understanding but also learning, as AI integrates the results of previous experiments to better solve the next experiment [1]. The current growth of artificial intelligence in healthcare is explained by the need to obtain useful and relevant information from the increasing amount of data available for each patient. Analysis of this large set of biological data, incomplete from human intelligence alone, requires the use of powerful computers. Big data are also present in medicine in the so-called -omics disciplines: Genomics, for example, evolved from genetics after it became fast and cheap to sequence entire genomes. Proteomics, which studies the molecular details of proteins, and metabolomics for metabolites are other examples. In medical imaging, image acquisition equipment is becoming more and more powerful (3 Tesla MRI, high-resolution CT), but the considerable amount of digital data collected is not sufficiently exploited by the human eye. Using the machine’s analytical capabilities, radiomics has become a supplementary tool to radiology: while the human eye performs a qualitative analysis of the image (shapes, sizes, enhancement profiles, density, etc.), the machine is able to extract and exploit a large number of quantitative features. Exploitation of this new information could, therefore, provide relevant information for physicians regarding diagnosis, prognosis, and treatment decisions [2]. Since 2012, this discipline has experienced exponential interest from the scientific community, as shown by the increasing number of publications on the subject (Figure 1) [3]. Indeed, radiomics could help us glean novel insights from medical images, which we can only infer by postulating that the pixels may be arranged in patterns that, once quantified, could signify an event of clinical relevance to both physician and patient. Further, the radiomics research carried out so far probably reflects only a small part of the information that can be provided by the analysis of such a large amount of data, limited by the computational capabilities of currently available graphics cards (GPUs: Graphic Processing Units). 

Radiomics fits perfectly with the current desire to precisely describe each pathology on an individual scale [4]. In oncology, particularly, the physician has numerous tools to accurately identify his patient’s tumor and will have even more in the future (Next-Generation Sequencing, research on the specific expression of a key cellular receptor, etc.). This practice of precision medicine is naturally suitable for the use of holistic models, integrating radiomics features as well as demographic, clinical, biological data, etc. [2].

### Characterization of Intra-Tumoral Heterogeneity

It is now well established that a tumor is not composed of a homogeneous set of the same cancer cells but rather of the coexistence of multiple sub-clonal populations, both morphologically and in terms of their molecular expression. We, therefore, speak of intra-tumoral heterogeneity, in contrast to inter-tumoral heterogeneity, which refers to the variations that exist between tumors of different individuals [5]. It has been found that the greater the heterogeneity, the more guarded the prognosis [6]. Friemel et al., in 2015, analyzed 23 HCC surgical specimens and showed that 83% of them were heterogeneous, both in terms of morphology (architecture, intracellular fat or bile content) as well as molecular expression or mutational profiles [7]. Nevertheless, a proper assessment of this heterogeneity is only possible with a full tumor tissue sample and is not achievable with a biopsy alone. Yet, it represents a challenge for precision medicine, with direct therapeutic consequences; if a mutation identified as predominant in the biopsy sample is minor in the whole tumor, the specific treatment of this mutation will not be effective, especially if we consider that another mutation, predominant in the tumor but not identified by the biopsy, is actually inhibiting the treatment that was believed to be specific. Aerts et al. performed the first radiomics study in 2012, involving the analysis of lung and upper aerodigestive tract cancer CT scans. In addition to showing that radiomics data can be associated with prognostic information, they confirmed that certain radiomics features varied according to different gene expression profiles within the same tumor and were, thus, able to capture intra-tumoral heterogeneity [8].

Unlike a biopsy, which only allows for the exploration of a sample, radiomics performs an analysis of the whole tumor. It, therefore, seems to be a useful tool for appreciating the degree of heterogeneity present within a tumor, particularly hepatocellular carcinoma. In addition, radiomics offers the advantage over biopsy of being non-invasive and easily repeatable over time, at different stages of the management of a cancer patient. Nevertheless, it has been shown that numerous radiomics features are dependent on tumor volume, which are, therefore, not usable in longitudinal studies, during which tumor volumes will change with the course of the disease or following therapy [9].

Beyond human clinical studies, patient-derived tumor xenografts (PDXs), which are cancer models made by the implementation of cells from a patient’s tumor into an immunodeficient mouse, recapitulate the heterogeneity of human tumors well [10], which can be operated by radiomics, as shown by Shoghi et al., who identified robust radiomics features from PDXs [11].

## 3. Conducting a Radiomics Study

A radiomics study consists of developing a retrospective predictive model of an event of interest from radiomics data extracted from a cohort of patients prior to the event occurring [2]. The first step of the study is to identify a cohort and define the event of interest, which must be relevant and have a significant impact on patient care. The next step is to select the imaging technique (CT, MRI, PET-CT, etc.) and to identify the Region Of Interest (ROI) where the radiomics data will be extracted. Typically, the ROI corresponds to the primary tumor, but it can sometimes be its metastasis or, more exceptionally, a healthy tissue area surrounding the tumor, the peritumoral area, where it is supposed that cellular events participating in the aggressiveness of the tumor’s growth can occur. Radiomics analysis of this peritumoral area could, thus, provide a good indication of tumor aggressiveness. 

The tumor must, therefore, be segmented for each patient, using either a semi-automatic or manual method. However, the manual approach is more time-consuming and less reproducible [2]. Software that performs semi-automatic segmentation identifies the margins of the tumor in an identical manner, regardless of the tumor analyzed, in contrast to human operators who may judge the tumor limits differently. Nevertheless, the semi-automatic method still requires human intervention to adjust the tumor margins when the boundaries are too blurry [12]. In the case of manual segmentation, delimitation of each tumor can be performed several times by the same operator or by different operators to evaluate the inter- and intra-observer reproducibility of the extracted features. However, the delineation between tumor lesions and normal liver can be made difficult by the heterogeneity of HCC lesions, in particular in the case of radiomics on PET-CT due to the variable Fluorodeoxyglucose (FDG) uptake by HCC. Blanc-Durand et al., thus, proposed to analyze the entire liver (including the tumor within the liver) when performing a radiomics study by PET-CT. Using, mainly, two textural features (strength and variance), they developed a predictive score for progression-free survival and overall survival after transarterial Yttrium90 radioembolization (90Y-TARE) for unresectable HCC. Even after stratification by Barcelona Clinic Liver Cancer (BCLC) stage and tumor size, the score differentiated a low-risk group (median progression-free survival (PFS) at 11.4 months (95% CI: 6.3–16.5 months), median overall survival (OS) at 20.3 months (95% CI: 5.7–35 months)) from a high-risk recurrence group (median PFS at 4.0 months (95% CI: 2.3–5.7 months), median OS at 7.7 months (95% CI: 6.0–9.5 months). The authors suggested that a radiomics analysis of the whole liver could, through a balance of normal liver tissue and tumor burden, provide prognostic information for HCC patients treated with 90Y-TARE [13]. 

These features extracted from the ROI can be classified into different categories. The first-order features globally describe the gray-level intensity distribution of the pixels. This includes data from the histogram, which represents the pixel frequency (on the ordinate) as a function of their intensity (on the abscissa): average and maximum intensity, spread, dispersion, asymmetry, etc. Second-order features, most frequently used in radiomics, describe the distribution of pixel intensity, not globally as first-order features but by considering the intensity of each pixel and its relation with its neighboring pixels. Figure 2 illustrates two of the most used second-order features: the Gray-Level Co-occurrence Matrix (GLCM), which describes the frequency of side-by-side appearance of two pixels of different gray levels, and the Gray-Level Run-Length Matrix (GLRLM), which describes the frequency of side-by-side appearance of two or more pixels of the same gray levels [12]. 

The artificial intelligence algorithm then analyzes these numerous parameters and identifies those that are correlated to the occurrence of the event of interest. Once identified, they are grouped to form a score or a predictive model. This model can then be improved to enhance its predictive power by adding non-radiomics data, such as clinical, demographic, or biological data [14]. Two cohorts can be constituted in order to test and validate the model, which can be evaluated using the Area Under Receiver Operating Characteristic (ROC) curve (AUC).

This validation step is essential and opens the possibility of generalizing the model created to current clinical practice [3].

Figure 3 provides a diagram of the different stages of a radiomics study.

## 4. Limitations

Although radiomics appears to be a promising tool in healthcare, several limitations remain that hinder its widespread use in routine clinical practice. Models created in each study cannot currently be compared since they do not follow a standardized protocol for implementation. Imaging techniques (CT, MRI, PET-CT) are generally different between studies: type of machine, voltage, acquisition time after contrast injection, etc.

To overcome this problem, Guiot et al. recommended that each author proposing a radiomics model should provide a precise summary of the characteristics of the study: acquisition process, ROI segmentation method, type of image processing, data filtration device, etc. Ideally, the code translating the modeling methodology used should be available [3]. In order to provide a better framework for practices, Lambin et al. proposed a quality score for radiomics studies, the Radiomic Quality Score [2]. It is based on 16 criteria for evaluating the quality and reproducibility of a radiomics study and could allow for a standardization of practices to make radiomics a tool for routine use.

More generally, artificial intelligence is currently limited by the excessive heterogeneity in patient data, as algorithms are currently unable to compare data from different imaging systems (MRI and CT scans, for example) and, more broadly, data from different disciplines (radiological and biological).

Moreover, access to radiomics is currently limited: the software is specialized and difficult to use. LIFEx, for example, a software program available in opensource, requires a trained operator.

## 5. Radiomics in Practice: Example of Hepatocellular Carcinoma

Hepatocellular carcinoma is a frequent (782,000 new cases in 2012), severe (third leading cause of cancer death, with 746,000 deaths in 2012), and singular tumor, whose geographic distribution is closely related to risk factors: viral, alcoholic, and/or metabolic liver diseases, with or without cirrhosis. Non-invasive diagnosis is possible with dynamic imaging (MRI/CT) for nodules > 1 cm on cirrhotic liver, according to EASL/AASLD criteria [15,16]. After being disregarded for about 20 years, biopsy is once again strongly recommended for initial characterization of a suspected HCC lesion. However, the clinician may not always have access to it, and a single biopsy alone is not sufficient to consider the heterogeneity of the tumor and, consequently, its overall prognosis [17]. In addition, the clinician has numerous medical CT or MRI images whose analysis by radiomics can provide valuable information concerning diagnosis, prognosis, or choice of treatment (Table 1).

### 5.1. Radiomics for Histological Diagnosis

Radiomics can be used to predict the precise diagnosis of a tumor, especially liver tumors. In subjects without chronic liver disease, liver tumors with a large vascular contingent may be difficult to characterize. Epithelioid angiomyolipoma is a vascular tumor with a variable fat content that makes it difficult to distinguish from focal nodular hyperplasia or hepatocellular carcinoma, two other blood-rich hepatic masses. Liang et al. studied CT (*n* = 170) and MRI (*n* = 137) radiomics features of patients with suspected epithelioid angiomyolipoma tumors. By comparing these features with the definitive histological results from biopsy or surgery, they obtained a model that distinguishes epithelioid angiomyolipoma from other liver tumors, such as HCC or focal nodular hyperplasia. In the validation cohort, the model from CT had an AUC of 0.879 and of 0.925 from MRI. After adding age, sex, and maximum tumor diameter to the model, the AUC was 0.966 (CT) and 0.971 (MRI) [18].

In cirrhotic patients, hepatocellular carcinoma can be diagnosed in the case of a nodule greater than 1 cm with a typical enhancement pattern. For nodules with a diameter between 1 and 2 cm, these non-invasive criteria have a specificity and a positive predictive value of 100% but a sensitivity of 71% [17]. Zhong et al. proposed a model to better differentiate HCC from benign nodules when liver tumors are greater than or equal to 3 cm. It is a model that combines radiomics features with the LI-RADS radiological classification [19], offering strong diagnostic performance, with an AUC of 0.975, a sensitivity of 97.3%, and a specificity of 97.7%. Without the addition of the LI-RADS classification, the radiomics model alone had an AUC of 0.917, a sensitivity of 93.8%, and a specificity of 86.4%. The LIRADS classification alone (without the addition of the radiomics features) had an AUC of 0.898, a sensitivity of 93.8%, and a specificity of 81.8%. The combined model significantly improved specificity (*p* = 0.030) and positive predictive value (*p* = 0.031) but did not significantly increase sensitivity (*p* = 0.215) and negative predictive value (*p* = 0.188) compared to the LI-RADS classification alone, showing that the contribution of radiomics here is modest compared to previously validated non-artificial-intelligence-based classifications [20].

### 5.2. Radiomics for Prediction of HCC Outcome

Among curative HCC treatments, surgical resection, restricted to early stages without significant portal hypertension, offers a 70% survival rate at 5 years, with a 70% recurrence rate at 5 years [17]. Post-surgical recurrence is, therefore, a major problem that is difficult to predict. It can be better assessed by non-radiomics models, using standard variables of interest: gender, tumor size, multinodularity, ALBI score, and AFP level, allowing one to distinguish three risk groups: low (20%), moderate (42%), or high risk (65%) [21]. Regarding radiomics, Ji et al. proposed a model to predict postoperative HCC recurrence based on preoperative radiomics data from 470 patients (divided into a training cohort: 210 patients, an internal validation cohort: 107 patients, and an external validation cohort: 153 patients) after a median follow-up of 56 months. Again, this is a mixed predictive model, combining radiomics data with clinical data: presence or absence of cirrhosis, radiological data: regular or irregular tumor margins, and biological data: ALBI score and AFP level, to obtain a better predictive power, with a time-dependent AUC of 0.803. The model allowed for a classification of patients into three groups: low risk of recurrence (median time to recurrence 98.7 months), intermediate risk (median time to recurrence 28.3 months), and high risk (median time to recurrence 6.4 months). The model was significantly better than other widely used staging systems (BCLC, ERASL, HKLC, CLIP, AJCC TNM: *p* < 0.05 for all models) [14].

Microvascular invasion is one of the elements most correlated to the risk of postoperative recurrence of hepatocellular carcinoma [22]. Further, its presence could challenge the indication for liver transplantation for HCC [23]. In order to predict the risk of HCC recurrence after surgery or transplantation, there is, therefore, an interest in non-invasively detecting the microvascular invasion from the medical image before surgery. Specific MRI criteria with the use of a specific contrast product have been proposed but have not been implemented in routine practice [24]. By performing a radiomics analysis of the tumor and the peritumoral area of 160 patients, Feng et al. proposed a model capable of predicting the presence of microvascular invasion on a preoperative CT scan, which was attested by the histological analysis of the surgical specimen. The peritumoral area analyzed was a 1 cm-thick ring around the tumor, which is probably the area where microvascular invasion is most present and, thus, where radiomics features are most informative. Combining radiomics features from the tumor and peritumoral area, the predictive performance is good, with an area under the ROC curve of 0.83, a sensitivity of 90%, and a specificity of 75%. However, in this case, a specific contrast agent for hepatobiliary imaging, Gd-EOB-DTPA, was used for MRI, making HCC segmentation easier by sharpening margins on the image, thus affecting the reproducibility of the model when other contrast agents are used [25].

Additionally, to predict microvascular invasion, Shi H et al. proposed a model using radiomics data from both MRI and PET-CT images of 97 patients. The model performed well in distinguishing the presence of microvascular invasion (AUC of 0.917 (95% CI: 0.824–0.970)) from its absence (AUC at 0.771 (95% CI: 0.578–0.905)). After integration into the model of FDG-PET texture data, metabolic parameters (Standardized Uptake Value (SUV) max) and MRI-specific parameters (apparent diffusion coefficient (ADC), hypovascular pattern on the arterial phase of the injected MRI, non-regular tumor margins), the predictive performance improved, with AUCs of 0.996 (95% CI: 0.939–1.000) and 0.953 (95% CI: 0.883–1.000). The combined model was significantly better than the radiomics model alone on both the training and validation cohorts [26].

### 5.3. Radiomics for Prediction of HCC Response to Therapy

The BCLC classification has become the reference for HCC staging, with treatment options for each stage, considering the tumor burden as well as the underlying hepatocellular function [27]. Nevertheless, it is not uncommon to experience a borderline situation between two stages where the right choice of appropriate treatment is difficult to establish. The advantages and disadvantages of one treatment option or another seem to be equivalent. No subclassification [28] or score [29] clearly emerged to guide therapeutic decision making for intermediate-stage HCC. Although the latest version of BCLC classification incorporates morphology (diffuse or infiltrating), it can sometimes be difficult to choose between chemo-embolization and systemic therapy [27]. Thus, Sheen et al. proposed a radiomics model to predict non-response to chemo-embolization from the CT scans of 80 patients. Interestingly, the model is based on only two radiomics features, one corresponding to the size of homogeneous areas with low gray levels and the other the size of areas with less uniform gray levels; the risk of nonresponse to chemo-embolization was greatest when the first was low and the second high, thus reflecting the poor prognosis of tumor heterogeneity. By combining these two radiomics features (forming the Rad-score) with the T stage of the TNM classification, the predictive model had an AUC of 0.95, a sensitivity of 91%, and a specificity of 75%. The two radiomics features also showed good predictive performance when combined with the bilobar or nonbilobar character of the nodules (AUC 0.91) or with logAFP (Alpha-fetoprotein) (AUC 0.91) [30].

Immunotherapy, by inhibiting immune checkpoints, has changed the management of patients with certain cancers, including locally advanced or metastatic HCC. Nevertheless, a response to immunotherapy, which is highly effective when it occurs, cannot be obtained in all patients. There is, therefore, a challenge to predict which patients will respond and, thus, avoid potential side effects in others [31]. These equivocal results of immunotherapy could be explained by inter-individual variability in tumor lymphocyte infiltration, which implies that characterization of the tumor environment is an essential activity in the management of these systemic therapies [32]. However, it requires the ability to analyze the entire tumor tissue, which is not possible with locally advanced tumors. Chen et al. used radiomics to create a model to predict tumoral and peritumoral T-cell infiltration based on MRI and CT scans of 207 patients undergoing surgery for HCC and on the observed T-cell density based on the analysis of surgical resection specimens. A 1 cm-thick ring around the tumor was also analyzed, combined or not with tumoral radiomics data. In the validation cohort, the radiomics model using only tumoral features had an AUC of 0.640, a sensitivity of 53.9%, and a specificity of 61.4%. The radiomics model combining tumoral and peritumoral features had an AUC of 0.899, a sensitivity of 92.3%, and a specificity of 72.7%. These values were improved after the addition of biological data (AFP, Gamma-glutamyltransferase (GGT) and Aspartate transaminase (AST)) that allowed for the combined model to achieve an AUC of 0.934, a sensitivity of 84.6%, and a specificity of 84.1%. Radiomics could, thus, be used as a biomarker for predicting the response to these innovative treatments. However, its predictive performance is moderate when used alone, especially on the tumoral area only [33].

**Table 1 diagnostics-13-01303-t001:** Main characteristics of the studies presented.

Authors (Ref)	Objectives of the Study	Imaging Modality	Nmbr of Patients	Geographical Origin	Cause of Liver Disease	Performance of Radiomics Model Alone	Performance of Combined Model
Blanc-Durand et al. [13]	Predict survival after radio embolization	PET-CT	47	Europe	Alcool	NC	NC
Liang et al. [18]	Differentiate hypervascular liver tumors	CT MRI	CT: 170 MRI: 137	China	NC	AUC (CT): 0.879 AUC (MRI): 0.925	AUC (CT): 0.966 AUC (MRI): 0.971
Zhong et al. [20]	Differentiate between HCC and benign liver nodules	MRI	150	China	HBV	AUC: 0.917 Sens: 93.8% Spe: 86.4%	AUC: 0.975 Sens: 97.3% Spe: 97.7%
Ji et al. [14]	Predict postoperative recurrence	CT	470	China	NC	NC	tdAUC: 0.803
Feng et al. [25]	Predict microvascular invasion	MRI	160	China	HBV	AUC: 0.83 Sens: 90% Spe: 75%	Radiomics model only
Shi et al. [26]	Predicting microvascular invasion	MRI PET-CT	97	China	HBV	AUC (MVI+): 0.917 AUC (MVI−): 0.771	AUC (MVI+): 0.996 AUC (MVI−): 0.953
Sheen et al. [30]	Predict response to chemo-embolization	CT	80	Korea	HBV	NC	AUC: 0.95 Sens: 0.91 Spe: 0.75
Chen et al. [33]	Predict response to immunotherapy	MRI	207	China	HBV	AUC: 0.640 Sens: 53.9% Spe: 61.4%	AUC: 0.934 Sens: 84.6% Spe: 84.1%

Abbreviation: AUC = Area Under Curve, tdAUC = time dependent AUC, CT = Computed Tomography, HBV = Hepatitis B Virus, MRI = Magnetic Resonance Imaging, MVI = Microvascular Invasion, NC = not communicated, PET-CT = Positron emission tomography–computed tomography, Sens = Sensitivity, Spe = Specificity.

## 6. Conclusions

In healthcare, as in other fields, technological advances have drastically increased the amount of available data. Medical imaging has been impacted by the new challenge of using “big data”, which the human brain is incapable of processing exhaustively. The emergence of “artificial intelligence” [34], which can analyze high-throughput quantitative features from medical images, has resulted in the rise of radiomics. These advances provide new information on patients and their pathologies and allow for a better characterization and understanding of the various phenotypic presentations. A single cancer may have different characteristics from one individual to another, with a direct therapeutic impact. Moreover, within a cancerous tumor, a certain degree of heterogeneity can be found, related to the prognosis, but not systematically quantifiable by a single biopsy, which does not reflect the entire tumor. The need, therefore, arises for new biomarkers, such as radiomics, to better understand each phenotype and to achieve precision medicine.

Hepatocellular carcinoma is a complex and heterogeneous malignancy. Thus, the diagnosis is sometimes difficult and the prognosis hardly predictable. Further, the ever-increasing range of therapeutic options can make treatment choices even more challenging. Studies conducted on this topic illustrate what radiomics can contribute by guiding the physician from diagnosis to therapeutics. The models proposed in the literature are often mixed, with clinical or biological data added to the radiomics features, illustrating the fact that the exploitation of radiomics features alone is currently insufficient.

Moreover, a harmonization of radiomics practices is still required to make them reproducible and generalizable, allowing for radiomics use in routine clinical practice. It could then become a new low-cost, non-invasive biomarker that can be followed up over time, providing valuable information to improve patient management.

## Figures and Tables

**Figure 1 diagnostics-13-01303-f001:**
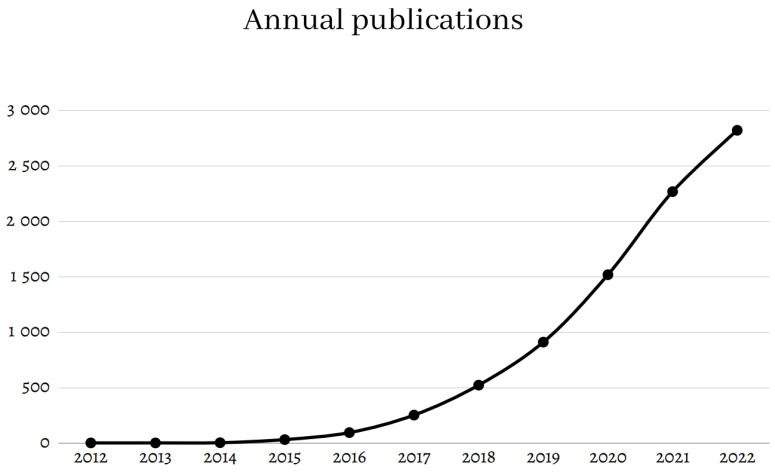
Evolution of the number of annual publications related to radiomics over the past 10 years. Data obtained from PubMed with the keyword “radiomics” in February 2023.

**Figure 2 diagnostics-13-01303-f002:**
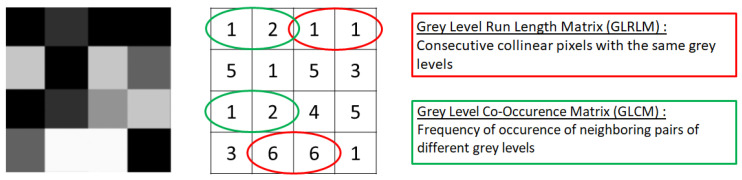
Example of two second-order radiomics features, Gray-Level Run-Length Matrix (GLRLM) and Gray-Level Co-occurrence Matrix (GLCM).

**Figure 3 diagnostics-13-01303-f003:**
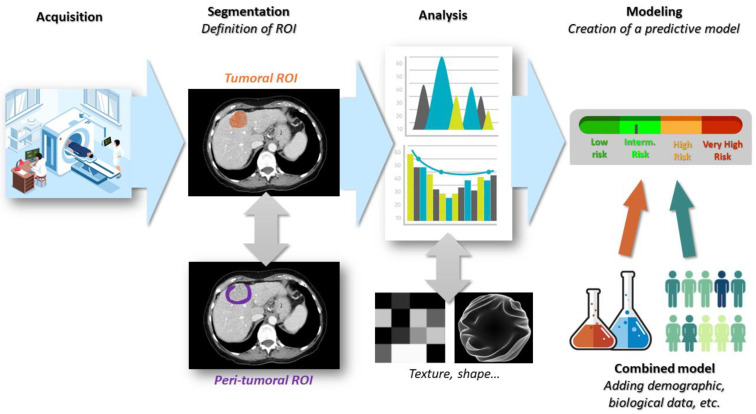
Process of a radiomics study. After image acquisition, the ROI (Region Of Interest) is segmented and features are extracted and analyzed to create the model, which can be combined with biological, demographic data, etc.

## Data Availability

Not applicable.

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
