# Peer review of "Radiomics, a Promising New Discipline: Example of Hepatocellular Carcinoma"

_diagnostics, 2023, doi:10.3390/diagnostics13071303_

Round 1
Reviewer 1 Report
AI and radiomics are hot topics currently. Lévi-Strauss's manuscript is written well and soundly. Some minor problems should be solved.
[minor problem]
1. The specific terms should be used in the full names instead of abbreviations in the first place, such as CT, MRI, PET (line 39), NGS (line 71), ROC, AUC (line 155), et al. The full name could be avoided in the following places, such as hepatocellular carcinoma (line 187).
2. The abbreviations used in Figure 2 are different from figure legends (no "matrix" in the figure; GLRL vs GLRLM)
3. All abbreviations should be explained in Table 1: TDM, IRM, PET-CT, CT, MRI, HBV, sens, spe, MVI+, MVI-
Author Response
Point 1: Specific terms have been used in the full names instead of abbreviations in the first place. Full name have been avoided in the following places.
Point 2: Abbreviations used in Figure 2 have been modified.
Point 3: Abbreviations in Table 1 have been all explained
Reviewer 2 Report
The emergence of "artificial intelligence", which can analyze high-throughput quantitative features from medical image, has resulted in the rise of radiomics. These advances provide new information on patients and their pathologies and allow for a better characterization and understanding of the various phenotypic presentations. A single cancer may have different characteristics from one individual to another, with a direct therapeutic impact. Moreover, within a cancerous tumor, there is a certain degree of heterogeneity, related to the prognosis, but not systematically quantifiable by a single biopsy, which does not reflect the entire tumor. There is therefore a need for new biomarkers, such as radiomics, to better understand each phenotype and to achieve precision medicine.
The paper need to clear the following queries:
1. What is the main question addressed by the research? 2. Do you consider the topic original or relevant in the field? Does it address a specific gap in the field? 3. What does it add to the subject area compared with other published material? 4. What specific improvements should the authors consider regarding the methodology? What further controls should be considered? 5. Are the conclusions consistent with the evidence and arguments presented and do they address the main question posed? 6. The readability needs to be improved, especially in the table part. It is difficult to match with the text. Many errors exist in descriptions of model performance. In general, I recommend the authors rethink the model design and seriously revise the paper before re-submission. https://doi.org/10.1016/j.ebiom.2020.102963 https://doi.org/10.1007/978-3-030-27272-2_14 https://doi.org/10.1007/s00259-021-05489-8 7. Presentation quality: Lack of professionalism and quality in presenting the block diagrams. It is suggested to use Lucidchart or TikzPlot to create block diagrams, even for graphs, plots, and bar charts. 8. Research question are not very clear from the studies. 9. the contribution is not clear.Author Response
First, I would like to thank you for your review of this manuscript.
This paper is intended to be a global and not detailed presentation of radiomics. It is aimed to a reader who has little or no knowledge of the subject and who is interested in the different possibilities of using artificial intelligence in health. Its aim is both to underline the interest of this emerging field but also to criticize its results which are sometimes insufficient when predictive models do not add non-radiomic data. This paper is not intended for an experienced audience in radiomics and does not address a specific gap in the field.
According to your recommendations:
-English has been revised.
-The readability has been improved and Table 1 has been detailed.
-The curve diagram Fig 1 has been redone with better software.
-https://doi.org/10.1016/j.ebiom.2020.102963 and https://doi.org/10.1007/s00259-021-05489-8 have been cited in the paper in the section “characterization of intra-tumoral heterogeneity”.
Best regards.
Reviewer 3 Report
This is a review well written article which can potentially increase the understanding of machine learning for entry level researchers. The era of digitization had in fact allows us to manipulate dimensions far more an eye can interpret.
1. include a citation at line 263
2. Line 165, change limits to limitations
3. Suggest a SWOT like analysis table for imporve the delivery on the message.
4. Spelling mistake [Author] in page 8
5. We recommend to include several articles in conclusion (Line 319)
https://www.nature.com/articles/s42003-020-01262-z
https://onlinelibrary.wiley.com/doi/full/10.1002/eng2.12383
Author Response
Point 1: The citation is included in the line 288 at the end of the paragraph.
Point 2: Limits is changed to limitations in line 179.
Point 4: Spelling mistake is corriged in page 8
Point 5: We included the nature paper in conclusion (line 342)
https://www.nature.com/articles/s42003-020-01262-z
Best regards
Round 2
Reviewer 2 Report
Author have revised the paper.